# Ecological niche modeling for surveillance of foot-and-mouth disease in South Asia

Umanga Gunasekera[1]*, Moh A. Alkhamis[2], Sumathy Puvanendiran[3], Moumita Das[1], Pradeep L. Kumarawadu[4], Munawar Sultana[5], M. Anwar Hossain[5], Jonathan Arzt[6], Andres Perez[1]

1 Department of Veterinary Population Medicine, College of Veterinary Medicine, University of Minnesota, St Paul, Minnesota, United States of America, 2 Department of Epidemiology and Biostatistics, Faculty of Public Health, Kuwait University, Kuwait City, Kuwait, 3 Department of Animal Production and Health, Veterinary Research Institute, Peradeniya, Sri Lanka, 4 Department of Animal Production and Health, Animal Health Division, Peradeniya, Sri Lanka, 5 Department of Microbiology, University of Dhaka, Dhaka, Bangladesh, 6 Foreign Animal Disease Research Unit, USDA-ARS, Plum Island Animal Disease Center, Southhold, New York, United States of America

* gunas015@umn.edu

## Abstract

Control of transboundary diseases at a regional level is commended over the country level due to its inherent complexities. World Organization for Animal Health (WOAH) has established different zones worldwide to control such contagious diseases as foot-and-mouth disease (FMD). Controlling FMD is difficult because of the complicated connection between FMD risk factors, and the deficits of surveillance activities in countries. We used an ecological niche model (ENM) that accounts for the under-reporting of outbreaks to determine FMD risk and risk factors in South Asian countries India, Bangladesh, and Sri Lanka. Centered on known outbreak information, we predicted high-risk areas using similar regional ecological features. Using a multi-algorithm machine-learning ensemble that includes random forest, support vector, and gradient boosting, 15 predictive variables (i.e., livestock densities, land cover, and climate), 660 FMD outbreaks from 13 years (2009–2022) in the region including the outbreaks from India, Bangladesh, and Sri Lanka we identified that Sri Lanka and Bangladesh appeared to have low to medium outbreak risk in the range of 0.04 to 0.55. India was used to fit the model. The machine learning models demonstrated high predictive performance (accuracy >0.87) through cross-validation. Production systems, isothermality, cattle density (per Km$^2$), and mean diurnal range was identified as the most important predictors of FMD outbreaks. These models help to determine FMD low-risk areas to minimize FMD surveillance activities and high-risk areas to focus on performing additional confirmatory testing, and improve surveillance in a regional context.

## Introduction

Foot-and-mouth Disease (FMD) is a transboundary animal disease caused by an RNA virus in the *picornavirdae* family. FMD is transmitted between animals via contact, animal products, and contaminated fomites including humans [1]. The disease is a threat to global

**Data availability statement:** Outbreak data used in this study are from three countries obtained with collaboration of listed coauthors in each country. Listed below are the links to Sri Lanka and India government websites where the data are available. Sri Lanka: https://daph.gov.lk/downloads/daph-publications accessed on 3/13/2025 India: http://www.pdfmd.ernet.in/ accessed on 3/13/2025 For Bangladesh, no central FMD outbreak data repository is available. Data used here are published in the following link cited in the manuscript. Further information should be obtained by contacting the authors of that manuscript; Bangladesh: 'Epidemiological Surveillance and Mutational Pattern Analysis of Foot-and-Mouth Disease Outbreaks in Bangladesh during 2012–2021'. https://onlinelibrary-wiley-com.ezp3.lib.umn.edu/doi/10.1155/2023/8896572 accessed on 3/13/2025 Other data source links are included in the supplementary files including the outbreak locations we used in this study. Interested researchers can replicate the study findings by obtaining the data from the third party sources and following the information outlined in the Methods section.

**Funding:** This project was funded in part by a grant from the USDA:ARS. The funders had no role in study design, data collection and analysis, decision to publish, or preparation of the manuscript.

**Competing interests:** The authors have declared that no competing interests exist.

food security due to trade restrictions for affected countries. In the South Asia region, FMD affects livestock farmers that only raise a few animals as their livelihood. In the South Asia transboundary animal diseases coordination meeting in 2023, both Food and Agriculture (FAO) and the World Organization for Animal Health (WOAH) have indicated the importance of controlling FMD at regional levels such as in South Asia, South America, Africa, the Middle East, or South East Asia with collective actions and initiating discussions among countries. The recommendations are to share information related to strategic control plans, border movement monitoring, value chain analysis among the countries and have a regional vaccine calendar to synchronize vaccination. Except India which is in stage 4 of Progressive Control Pathway (PCP), other countries in South Asia are in the PCP stages 1, and 2, or progressing towards stage 2 including Sri Lanka, and Bangladesh (WOAH Report). Activities in stage 1 of PCP include identifying the livestock-marketing network, importation of animals, animal products, and animal movement among other aspects. It is expected that the countries in the stage 1 demonstrate commitment to regional FMD control.

Predicting FMD risk based on reported outbreak numbers is unreliable due to drawbacks in passive surveillance activities such as underreporting and late reporting. Historically, this limitation has been compensated via different analytical methods such as standardized incidence ratios, spatial methods such as kriging, kernel density [2], and spatial Bayesian methods [3,4]. Compared to spatial methods such as kriging [5] and sat scan analysis, which uses only outbreak data location and time [6] to determine FMD risk areas, ecological niche models (ENM) can accommodate and explore many highly correlated and complex risk factors to predict the spatial risk of FMD robustly. ENM use variables, such as temperature and precipitation, along with reported incidences to detect species abundance and predict outbreak occurrence [7]. One of the most commonly used ENM is maximum entropy species distribution. However, this model requires several assumptions such as representative sampling, and considers presence-only data. For the algorithm component, supervised machine learning methods based on decision trees and kernel function are recommended over this method using both presence and absent data [8]. Interpretable multi algorithm machine learning models require fewer statistical assumptions, are less sensitive to highly correlated variables and, therefore, overfitting, and can explore none-linear complex relationships between variables [9]. Also, Machine learning (ML) methods have been widely used to predict veterinary infectious diseases [10–14].

For FMD and other infectious diseases, there is no straight linear pattern that differentiate infected from the non-infected [15]. Machine learning approaches are capable of exploring these nonlinear interactions [16]. Different environmental risk factors such as the relative humidity, temperature are associated with FMD persistence in endemic countries [17]. To predict the FMD risk, location of outbreak occurrence data are combined with different environmental predictor's spatial distribution [16]. For FMD, ML has been used in studies in Thailand [18], South Africa [19], and China [20].

We hypothesize that various environmental, epidemiological, and demographic factors could serve as a proxy to predict FMD risk. The objectives of this study were to a) use interpretable supervised machine learning algorithms based on decision trees and support vector methods to develop risk prediction models based on environmental data for FMD in South Asia using empirical outbreak data from India, Sri Lanka, and Bangladesh. b) To identify major risk factors shaping FMD risk and c) compare predictive performance among different models used. These results will help Bangladesh and Sri Lanka to move forward in their PCPs for FMD, ultimately contributing to the control of the disease in the region.

## Methods

For this analysis, reported outbreak data were obtained from India, Bangladesh, and Sri Lanka from 2009 to 2022. An outbreak was defined as a group of epidemiologically linked cases (WOAH, Terrestrial code) during the considered period.

In Bangladesh, a systematic FMD outbreak surveillance system is not available. Therefore, it is possible that not all outbreaks are reported during the period. The outbreak data used in this study are from a field study conducted during 2012–2021 covering 32 different districts. Outbreak information are collected based on farmer's notification for serology testing. A total of 481 samples were collected from different outbreak. Sample collection was affected by the 2020–2021 COVID 19 outbreak [21]. In this study, we used sample collection location confirmed by laboratory testing affiliated district as the outbreak location. We did not consider the temporal aspect of data. Compared to data coming from Sri Lanka and India, data from Bangladesh is not complete.

For Sri Lanka, officially reported outbreak data were available from the Department of Animal Health, Sri Lanka. The respective veterinarians from nearly 256 veterinary ranges report outbreaks as a part of passive surveillance activities. A district in Sri Lanka consist of multiple veterinary ranges. Outbreak reporting system is a paper based monthly report sent to the head office. Total 369 outbreaks were reported during the period of 2014–2022 from different areas of the country. A reported outbreak may have one to many infected animals in an identified farm location. If one or more outbreak was reported at a VS range, a point location of the VS range was considered as positive and recorded for this study. These outbreaks are clinically identified initially with later confirmatory serotype diagnosis.

In India, veterinary authorities conduct both active and passive surveillance for FMD mainly focusing on passive surveillance in nearly 65000 administrative levels. Confirmatory diagnosis is carried out for serotyping in 27 FMD network laboratories and 2 national laboratories. A reported outbreak may have one to many infected animals in an identified location. The reporting system is paper based and a monthly report is submitted to the Department of Animal Husbandry. The outbreak location was considered up to the district level (i.e., if there are one or more outbreaks reported at village level, for the purpose of this analysis, the district was considered positive and a point location of the district was recorded). FMD outbreak data for India (n = 429) were available at the district level from a previous study for the years 2009–2020 [3].

For FMD Diagnostic testing, the countries follow the guidelines of 'Manual of Diagnostic Tests and Vaccines for Terrestrial Animals, 13th edition 2024", WOAH terrestrial code. Number of outbreaks from each country each year, is shown in the S1 Fig. Following the spatial analysis ML algorithm [10], to lower the training error of the model, duplicate occurrences of the same geographical location of outbreaks were removed during the considered period. From a total of 829 outbreaks, the final data set comprised of 660 outbreaks from the three countries. While outbreak data is considered in this study, the model predict suitable places where FMD outbreaks are most likely to occur where it was not reported or underreported considering other features as well.

For the predictors, we used FMD risk factors such as climate data [22,23], production systems [24–26] and livestock densities [23,27,28]. Because animal movement records are not available, the road network was incorporated as a proxy indicator of animal movement for trade and market [4,29]. Extensive farming practices are common in the region [30], and land cover includes land use (different types of forests, cropland, land that can be used for pastoralism, and wetlands) that is associated with extensive grazing. Therefore, the Normalized Difference Vegetation Index (NDVI) was considered. A summary of different predictors is shown in S1 Table.

Selected historical bioclimatic data are derived from monthly temperature and rainfall values representing annual trends and seasonality [31]. The dataset consists of 19 bioclimatic features related to temperature and precipitation. However, the mean temperature of the wettest quarter, the mean temperature of the driest quarter, the warmest quarter precipitation, and the coldest quarter precipitation were not included due to spatial artifacts. Data are average for the years 1970–2000 one for each month at 5-minute resolution.

Livestock densities of cattle, buffalo, goats, sheep, and pigs were obtained from the FAO, Gridded Livestock of the World database (GLW v4) database [32]. The latest data set available was from 2015. Global livestock distribution was expressed in the total number of animals per pixel (5 min of arc) at a km² census unit. The data are stored in geographic coordinates of decimal degrees based on the World Geodetic System spheroid of 1984 (WGS84).

Normalized difference vegetation index indicates the density of vegetation using sensor data obtained via satellites. Copernicus land monitoring service includes NDVI indexes at a 10-day interval at a global scale at a spatial resolution of about 300m from 2014 to June 2020 at WGS 1984 latitude longitude projection. We downloaded 10 raster layers in tiff format representing years 2014–2020 and obtained a combined raster layer using ArcGIS with the same cell resolution and the coordinate system.

The Cropland tiff file was obtained in tiff format from the FAO Crop Land – Global Land Cover Share Database. The database is created and validated by harmonizing different land cover databases. The cropland cover projection is rated at an accuracy of 94.9%. The dataset is in the raster format Geotif at WGS 1984 scale, representing the percentage density at a resolution of 30 Arc seconds.

Livestock production systems are different across the world. The livestock production system data are obtained from the FAO Global Livestock Production System at a 30 Arc second spatial resolution. In this database, production systems are considered as agro-pastoral systems, crop-livestock, and others.

The road network was considered a proxy for animal movement to livestock markets and slaughterhouses. Shape files are obtained from the Global Roads Inventory Project (GRIP) global roads database [33]. Roads are projected until the year 2050 using the existing open road databases from year 2000–2015. The data are stored as shape file in geographic coordinates of decimal degrees based on the World Geodetic System spheroid of 1984 (WGS84). Roads are displayed at 5 arc minute resolution under The Global Roads Inventory Project (GRIP) categories. Shape files is converted to a raster file using the feature to raster function in the ArcGIS. Road raster will be aggregated and resample to obtain the road density in R.

### Data processing

Raster package in R was used to convert different data containing files such as tif, and adf into one geodetic system; WGS84. Each variable was then cropped to the extent of South Asia. The relevant shape file was downloaded from the 'Natural Earth' (https://www.naturalearthdata.com/downloads/10m-cultural-vectors/). Because different variables have different spatial resolutions, all were aggregated and resampled to make it to the same spatial resolution of approximately 9 km².

As a requirement of the machine-learning pipeline random absent data points were created to make presence–absence data [9]. A preliminary analysis was run to optimize the variables, the spatial resolution and to determine the case-control ratio. Random absent data points were created maintaining a 1:1 case control ratio. Here the absence points characterize the environment in the study region [34]. Absence points were merged with the actual outbreak location data into one data frame and spatial probabilities were created to determine FMD risk areas.

Collinear variables were removed where the largest mean absolute correlation was greater than 0.9 based on the correlation matrix. The Boruta package in R was used to select statistically significant important features [35]. This package uses a Random Forest classifier. Once shadow features are created to account for random fluctuations, the Z score is used to determine important features. All the considered variables that were not correlated at a threshold of 0.9 were identified as significant when the Boruta package was applied.

The whole data set was divided into 80% training and 20% testing data for 10-fold cross-validation and to train the machine-learning algorithm. Cross-validation prevents overfitting of the data. The caret package in R is used for data partitioning [36].

## Model training and evaluation

For the selected features, we performed Random Forest (RF), Gradient Boosting (GB), and the Support Vector (SV) machine learning algorithms to create predictive models of FMD following Fountain Johns et al., 2019 using R package Caret [36].

Random Forrest (RF) and gradient boosting (GB) methods are based on decision trees. Decision trees provide classification and separate paths based on selected variables. The way decision trees are made is different for each method. The RF method is suitable when the data is sparse. In the RF method, variables from the bootstrap data (training data) are randomly allocated in decision trees. Trees are then randomly selected to test data (out-of-bag data) that was not used in creating trees [37,38]. The accuracy of the model is determined by test data. When GB makes decision trees, new trees are scaled, and made based on the errors of previous trees, and the size of the trees is restricted [39]. The support vector (SV) machine learning method uses a kernel function to classify data (outbreak vs no outbreak) at a higher dimension space based on a threshold value. The threshold accounts for the bias-variance tradeoff [40,41]. Ten-fold cross-validation was used to estimate model performance and compare different machine learning methods. Each model was run 10 times for 10 fold cross validation after the first run using training data.

Tenfold fold cross validation involve randomly dividing the data set in to k folds. The first set is considered as the validation set and remaining as training data to test the method. Here repeated 10 times on test data. K fold cross validation is expected to get a closer error to true error with repeated running. For the classification tree, we consider the number of misclassifications as the error. Performance was assessed for each run and the average was compared among the models. K fold cross validation reduce the bias variance tradeoff by including more variables and observations $((k-1)\,n/k)$ in each run [42]. Confusion matrices were used for each model cross-validation average to select the best model using the Caret package in R. Accuracy, Receiver Operator Characteristic (ROC), Mathews Correlation Coefficient (MCC), sensitivity (Se), and the specificity (Sp) of each model was calculated.

Accuracy measures the proportion of correctly identified observations. The sensitivity of the model is important to determine correctly identified FMD positive areas as FMD positive and the specificity of the model depicts correctly identified FMD negative areas as negative. The ROC helped to determine the optimal threshold ratio between the true positives (sensitivity) and the false positives (1-specificity) of confusion matrices that resulted from cross-validation. ROCs were built for each model separately. The area under the curve of ROC determines how well the model discriminates between FMD negative and positive areas and to determine that there is no data imbalance between the outcomes of interest [43].

The Mathews Correlation Coefficient (MCC) is a correlation coefficient that accounts for both true false positives and true negatives as a balanced measure ranging from −1 to +1. A coefficient closer to 1 indicates the higher predictive ability of all four categories of the

confusion matrix of each model. MCC is invariant to class changes and is considered a better measure compared to the F1 score of the confusion matrix [44].

Predicted FMD risk maps of India, Bangladesh and Sri Lanka were obtained for all three algorithms. RF predicted risk raster was exported to Arc GIS to obtain mean FMD risk values at higher administrative levels for Sri Lanka and Bangladesh. These values are presented as relative risks.

## Model interpretation

The predictive performance was high and comparatively similar across all models. For all the models, we considered feature importance, feature dependence, and the overall interactions for model interpretation.

Different variables are considered important in different models. It is important to determine the amount each predictor contributes to the model's accuracy. Model class reliance identifies a range of important variables across different well-performing models [45]. This is accomplished by changing each selected variable to understand the impact that change has on model performance. If it is significant, the variable is considered as important [46]. R package FeatureImp was used to create variable importance plots.

Centered Individual Conditional Expectation plots (cICE plots) calculated the individual effect of each variable on each response. These plots show individual effects and individual responses on each observation keeping all the other variables the same. cICE plots were created to show the top five variables of each model [47]. Partial dependency plots help to identify the relationships between the predictor's values and model predictions. This is an average estimate of a predictor if all data points assume the same feature value (a global estimate) on the outcome variable [47].

Interactions among variables increase with the increased number of predictor variables. Once important features are recognized, feature interaction strength is measured by the Friedman H statistic [48]. The interaction plots show the marginal impact of a variable on the predicted outcome.

## Results

Our analytical pipeline identified 15 of 24 variables as important predictors of FMD's spatial risk. Identified variables are production systems, Normalized Vegetation Index (NDVI), cattle, sheep, buffalo and, goat density, road density, cropland, maximum temperature in warmest month, isothermality, mean diurnal range, annual precipitation, precipitation dry month, precipitation seasonality, and precipitation in the wet months.

Based on the cross-validation approach, both random forest and gradient boosting models performed similarly. Therefore, we selected the RF for the subsequent predictions and interpretations (Table 1).

Predicted risk varied between 0.04 to 0.55 in Sri Lanka and Bangladesh, whereas India was predicted to be highest risk (mean value of 0.75), with most of the predicted risk throughout central India and some border areas remained high of the country (Fig 1).

Table 1. Summary results of cross-validation for different models.

| Model | ROC | Accuracy (%) | Specificity (%) | Sensitivity (%) | Mathews Correlation Coefficient |
|-------|-----|--------------|-----------------|-----------------|--------------------------------|
| RF | 94.45 ± 0.01 | 87.59 ± 0.36 | 88.28 ± 0.49 | 87.34 ± 0.53 | 0.75 ± 0.007 |
| GB | 93.21 ± 0.01 | 87.87 ± 0.31 | 88.91 ± 0.54 | 86.76 ± 0.43 | 0.75 ± 0.006 |
| SV | 89.51 ± 0.01 | 82.14 ± 0.55 | 82.03 ± 0.52 | 82.26 ± 0.83 | 0.64 ± 0.01 |

RF, random forest; GB, gradient boosting; SV, support vector.

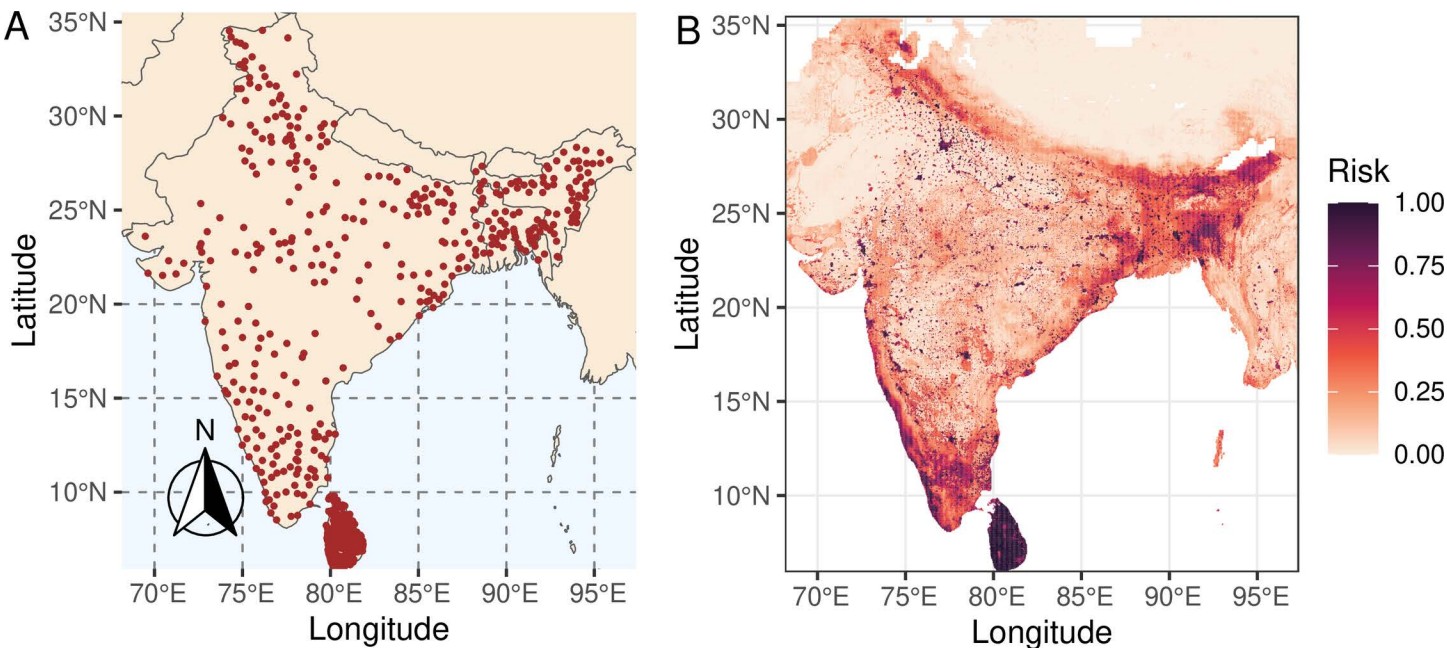

**Fig 1. a) Locations of reported outbreaks in India, Bangladesh, and Sri Lanka. b) Predicted high and low-risk areas probability distribution for FMD based on Random forest model.**

In Sri Lanka, both North and North Central provinces had a higher mean risk compared to the rest of the country (0.072 and 0.067). In Bangladesh, Barisal, Khulna and Chittagong divisions were identified to have a higher mean risk (0.682, 0.603 and 0.573) (Fig 2).

From each different model, production systems and isothermailty were identified as the most important features associated with the predicted FMD spatial risk (Fig 3). PD plots showed that the risk increased with production systems 5–10. Production systems 5–10 include rain-fed arid, humid, temperate, and mixed irrigated arid systems. The spatial risk of FMD increased and plateaued when isothemality =~ 35% (Fig 4). Isothermality indicates temperature variation. Interaction plots showed that the highest interaction between the production systems and isothermailty. Higher risk was predicted at higher values (Fig 5).

Considering the RF model, FMD risk increases with cattle density per $Km^2$, NDVI above 100. FMD risk was reduced with the mean diurnal temperature. The mean diurnal range is the difference between days' minimum and maximum temperature.

Fig 5 shows the interaction strength among the selected features and the interaction of production systems with the other features. These interactions are further explored by heat matrix (Fig 6).

Production systems followed by mean diurnal range, isothermailty, cropland, and wet month precipitation had the highest overall interaction with the other features. At higher isothermality (<60) and a lower mean diurnal range, the magnitude of the spatial risk is high with the 'production systems of mixed irrigated, urban, and other'. The interaction effect for dry month precipitation, cattle density, and NDVI in general remained high for the same production system.

## Discussion

Using machine learning algorithms, outbreak data spanning more than 10 years, and FMD risk factors, we identified areas at highest risk for FMD in Bangladesh, and Sri Lanka using

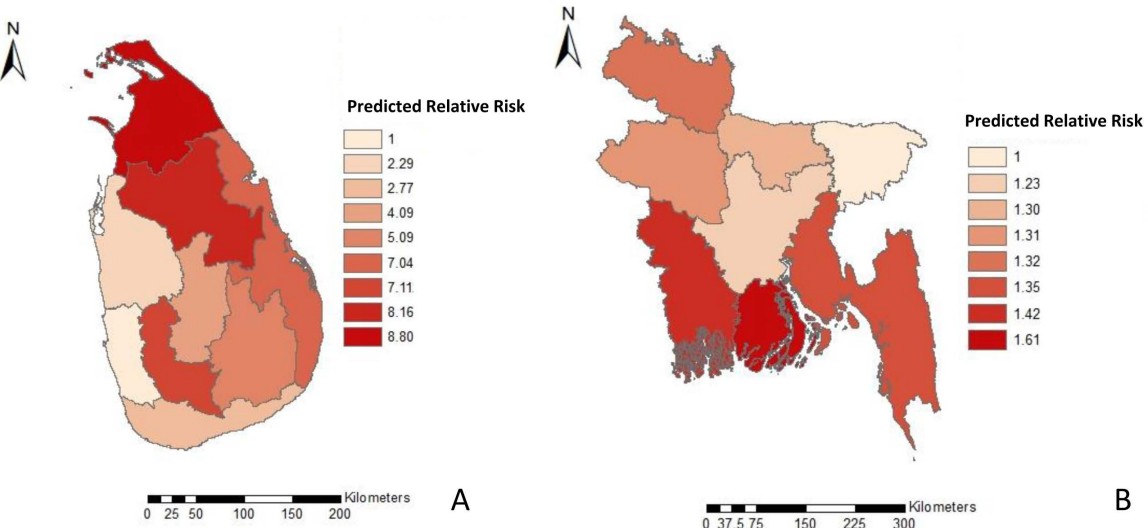

**Fig 2. Mean predicted relative risk for a) Sri Lanka and b) Bangladesh with the Random Forest algorithm.** Darker red colors indicate high-risk areas. Note: Size of the countries are not proportionate to each other.

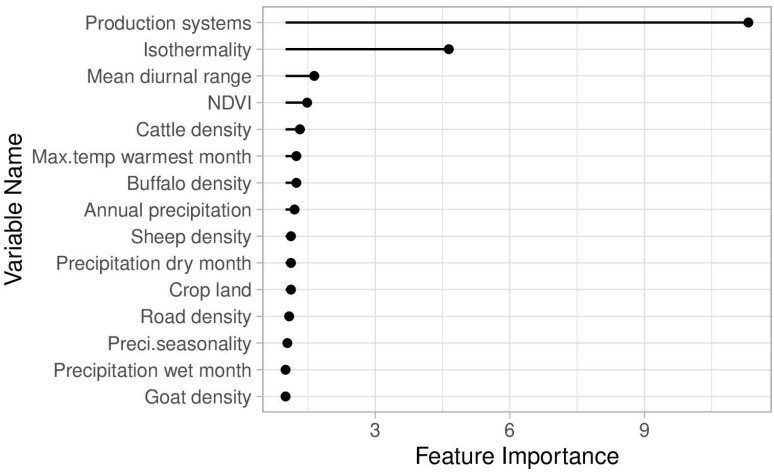

**Fig 3. Random Forest feature importance plot.**

India to fit the model. Feature importance plots identified the production systems of livestock management and the isothermality as the major risk factors shaping FMD risk in the region, followed by the NDVI, cattle density and the mean diurnal range. All of our selected ML algorithms (i.e., RF, GB, SV) showed high predictive performance. Our main result concluded that Bangladesh and Sri Lanka had lower risk for FMD outbreaks (i.e., p = 0.04 to 0.55) compared to India (p = 0.75).

India predicted high risk when environmental predictors were considered (0.75 mean risk). India is progressing in the PCP Pathway to stage 4 with FMD control program (FMD CP) officially endorsed by WOAH [49]. The country had achieved several milestones related FMD control program concerning vaccination, and surveillance of outbreaks. At stage 3 of the PCP, the objective of India was to do zonal compartments to enhance vaccination for progressive

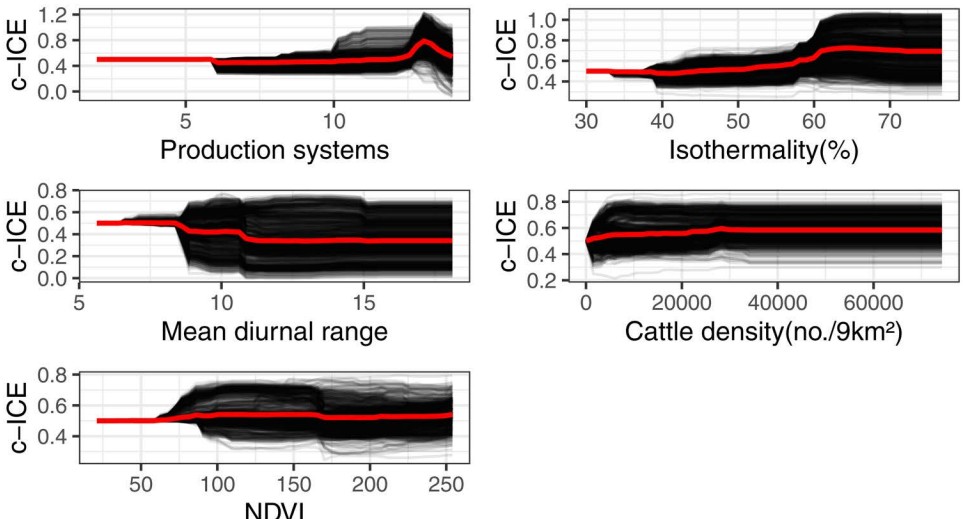

**Fig 4. Centered ICE (classification error loss) feature importance plots from Random Forest.**

reduction of the FMDV from the country. These activities are continued in stage 4. Despite control measures, FMD risk areas are still spread across India. In this model, we considered only environment-related variables for predictions and outbreak data from India until 2020. Therefore, it may not reflect the recent advances the country has made with the FMD control with the vaccination program [49].

Our previous spatial temporal Bayesian analysis in India that accounted for vaccination showed that border areas were at high risk compared to the central part of the country [3]. Findings from this study are compatible with that study, as comparatively high-risk areas are identified in the peripheral parts of the country.

Phylogeography studies have shown that the viruses originating from South Asia would move toward Southeast Asia and west towards the Middle East [50,51]. This is assumed because of illegal animal transport through international borders, and low vaccination coverage among other reasons [52].

Both Bangladesh and Sri Lanka were identified as low-risk compared to India. Both countries are in the PCP stage 1 (3rd SAARC Roadmap Meeting on the Foot-and-Mouth Disease Progressive Control Pathway (FMD-PCP). In Sri Lanka Northern Province bordering India and in Bangladesh, Barisal, Khulna and Chittagong divisions that border India are identified as high risk for FMD compared to the other areas. The eastern part of Bangladesh was identified as a high-risk area in a previous publication [22]. In Sri Lanka, the northern and eastern parts are identified as high-risk compared to the rest of the country [53]. These findings are compatible with high risk areas identified by our ML models. Collaborating to implement restrictions on illegal animal movement across international borders is beneficial to prevent FMD and other transboundary diseases. A requirement of PCP stage 2 is initiating a legal framework to ensure movement control and surveillance at the borders.

The association between risk factors for a transboundary disease like FMD is complex [54]. Use of logistic regression to determine risk factors does not show good accuracy when there are complex relationships among variables [41]. As identified from the RF algorithm there are considerable interactions among the identified major risk factors production systems with climate related features. The best-performing model identified among the different methods we tested here was Random Forest, which performed better compared to the regression model.

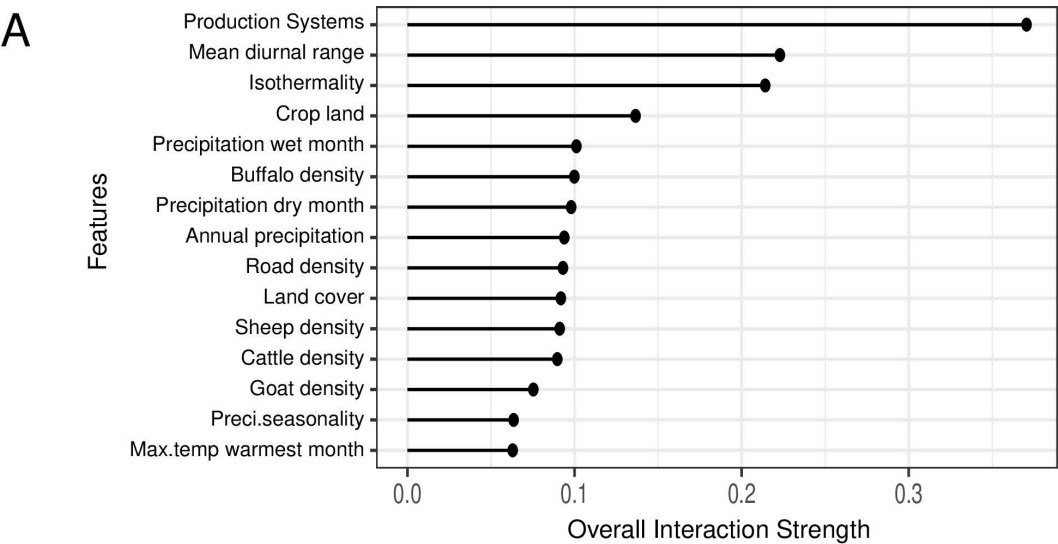

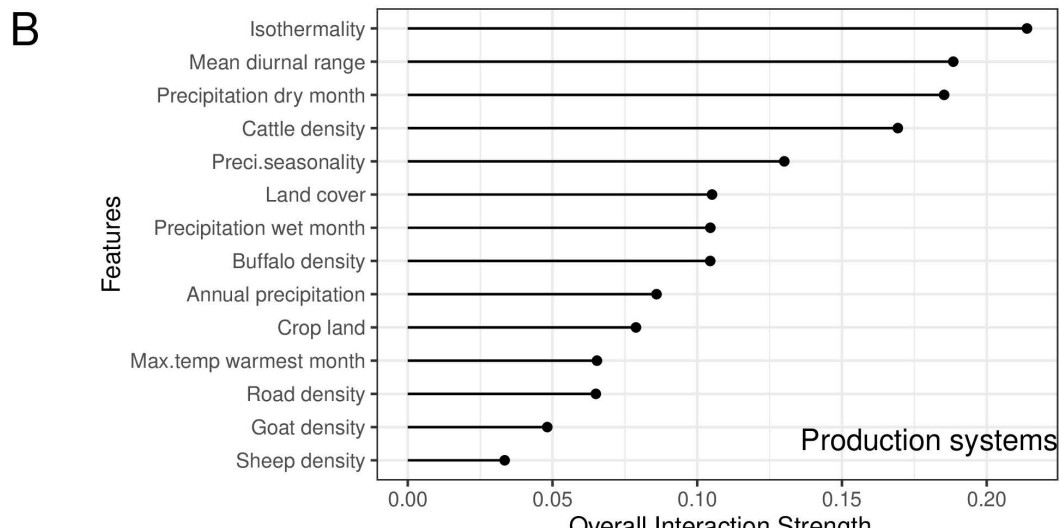

**Fig 5. A) Feature interaction plots of all the variables, production systems and mean diurnal range were identified with highest interaction.** B) Overall interaction strength of production systems with the other features. Isothermality and mean diurnal range has the highest interaction with the mean diurnal range.

Random forest and the other ML models we used do not provide a coefficient or causality but account for the complex association between risk factors via correlation [55]. In general, these associations are nonlinear. We considered different risk factors identified from the best performing RF algorithm production systems, isothemality, mean diurnal range, NDVI, and cattle density for discussion.

South Asia is population dense with 2.04 billion people (25% of the world population) (World Meter). According to the Commission on Global Poverty, World Bank, 34% of the world's extreme poor are in this region. The livestock is reared to provide emergency income and to utilize crop residues. The livestock management systems are such that, every household may have two to three animals. The productivity is low in this system compared to the other

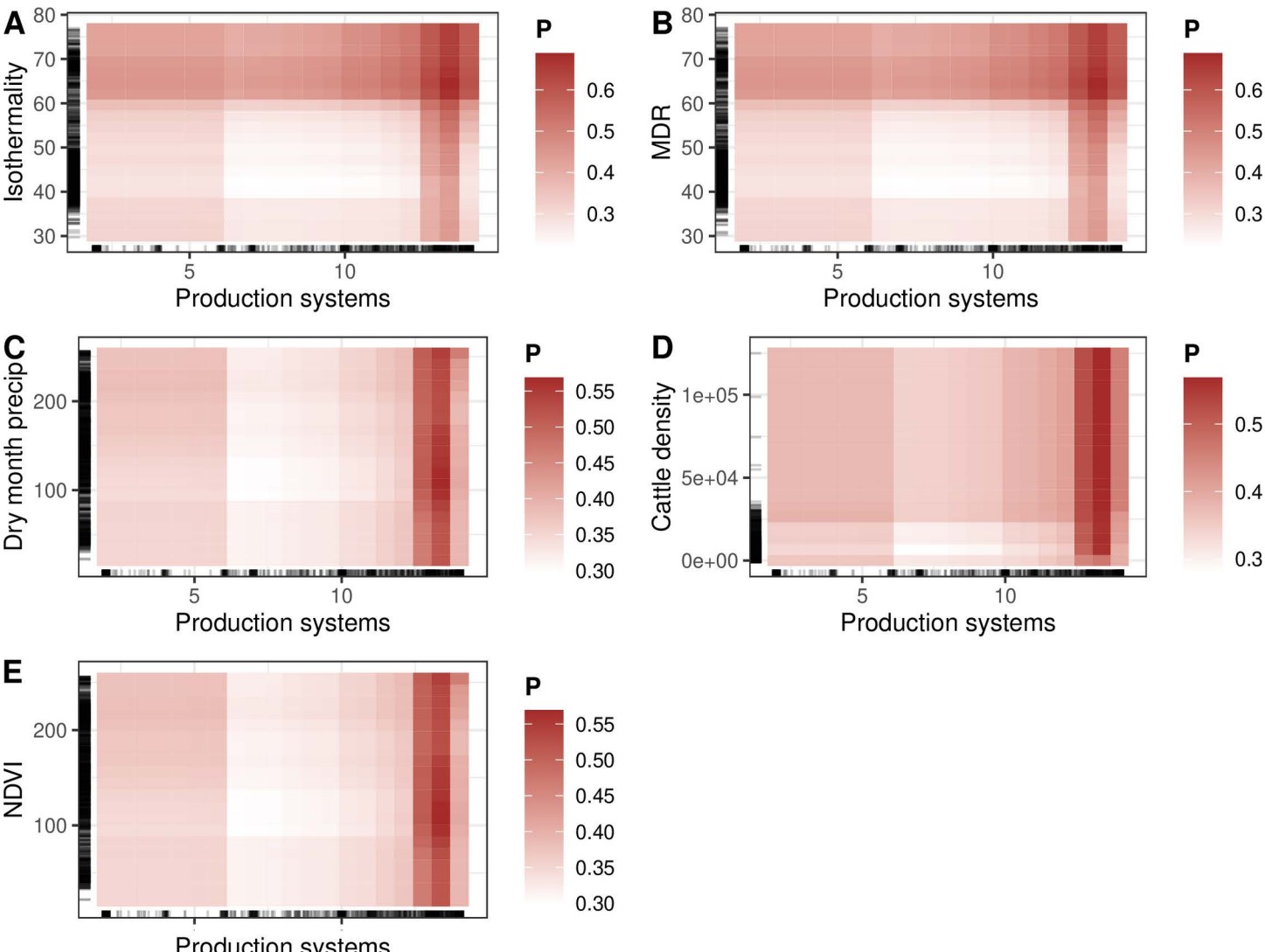

**Fig 6. Feature interaction heat matrix plots of A) Isothermality, B) MDR-mean diurnal range, C) Dry month precipitation, D) cattle density, E) Normalized vegetation index, with production systems.**

management systems [30]. The identified FMD risk factors in the area revolve around this system that is far from intensive management and therefore lacks bio-security measures.

FAO global livestock production systems that we used in this analysis account for human population density and these different types of management systems. Here, it was identified that FMD risk increases with mixed irrigated and rain-fed arid systems, as well as urban and other. Mixed irrigated systems are classified as extensive livestock management systems where 10% of production comes from non-livestock land use (irrigated systems) and where crop by-products are fed to animals. Rain-fed farming systems are classified where industrial crop activities are higher but livestock are also present. The rain-fed arid system is identified as a dispersed system with very low productivity due to extreme weather conditions. With high human population density, livestock is typical in urban areas as well. Normalized vegetation index (NDVI) was another risk factor identified by different models. A higher and moderate vegetation index is associated with higher FMD risk. This could be due to extensive grazing

management practices of the livestock where animals are sent to graze in forest and pasture lands.

Weather patterns are also associated with FMD risk such as isothermality, mean diurnal range, and precipitation. Isothermality is associated with the degree to which temperature varies throughout the year compared to the annual temperature variation [56]. The mean diurnal range is the difference between days' minimum and maximum temperature. This value has been decreasing with climate change [57]. According to the cICE plot, a mean diurnal range decrease is associated with high FMD risk. Precipitation seasonality is directly associated with monthly rainfall and FMD risk due to grazing patterns followed in the region where animals are moved to different grazing areas and the virus survives during dry and cool weather right after the monsoon rainy season [22,53,58]. Identification of this pattern is important to decide on when to implement control measures such as vaccination and movement restrictions, before animals get exposed to the virus.

FMD risk increases with cattle density. Cattle are the main livestock species infected by FMD in the region [22,52]. Controlling cattle movement is encouraged but is hard with existing management practices. In Sri Lanka, animal movement and communal grazing were identified as the major risk factors [53]. In stage 1 of PCP, it is important to identify FMD distribution in the country. Identifying areas with an interaction among the aforesaid variables that predispose to FMD is important to focus on to carry out risk-based surveillance activities.

An advantage of the ML approach we followed is the ability to untangle complex interactions. This advantage is lacking in linear models like logistic regression, where attempts to fit interaction terms usually lead to uninterpretable results (i.e., lack of epidemiological plausibility) and overfitting. Here, we identified production systems as the most overall interacting feature on one side and isothermality, mean diurnal range and cattle density features on the other (Fig 6). Production systems are inherently associated with the climatic variables such as seasonality, rainfall for both livestock production in the region [59]. Since no studies are available related to this exact production system, considering production systems infers farm density, there are evidence from simulation studies that high cattle densities will results in FMD outbreaks irrespective of farm density but farm density is associated with larger epidemics only in the presence of high cattle density [60].

In this analysis, decision tree models performed better than SV models. Another advantage of an RF model is that it can reduce the chances of overfitting using out-of-bag error data for internal validation [37]. Random forest is considered best performing with a larger number of variables than the number of observations compared to other ML methods. Here, we have fewer observations compared to many different machine-learning models used in other fields, e.g., gene editing and gene expression [61], and patient outcome prediction [62]. The caret package selected the number of predictors and optimal branching for RF.

It is identified with RF, that there can be selection bias when the scale of measurement or the number of categories varies across variables [63]. Here we selected our scale of measurement similar across many variables by aggregating and resampling before further analysis and did not include any categorical predictor variables. Even if there is selection bias, it does not affect the selection of important variables [37]. For variable importance measure (VIM), we use the permutation method [46] based on prediction accuracy instead of the Gini impurity index which is based on the splitting criteria, as VIM is based on prediction accuracy and is considered better than Gini impurity index [37].

There are a number of limitations in our work. The differences in surveillance capacities of the three countries that lead to data inconsistencies are a limitation when comparing the countries for FMD risk. India is having higher diagnostic capabilities compared to both Sri Lanka and Bangladesh. Since we are predicting the suitability for all types of occurrences

(reported outbreaks and laboratory confirmed outbreaks via antibody detection) using an ecological model, ecological fallacy may affect the results. To a degree, this is compensated as we regarded only the spatial aspect of data. Our study encourage authorities in all the three countries to improve the quality of FMD surveillance particularly Bangladesh where data was not available from a central source.

Since feature selection via Boruta was performed before splitting the data into training and test sets there is a potential of data leakage [64]. Such impact should be minimal as our data set is smaller and the model evaluation results were consistent across different validation methods. In an ecological niche modeling environment, it is important to select, a scale that fits the biology of the disease that is modeled [7]. We have used 5 minute arc is a pixel resolution of about 9 km [31]. FMD is transmitted at a range of 50 Km to 200 Km [65]. There could be limitations to interpolated climatic data [66]. Since this model is not at a fine scale, detailed interpretations cannot be made based on these findings. Here we did not consider the temporal aspect of FMD outbreaks but FMD has a strong temporal component that results in the specific epidemic curve, which indicates this identified high-risk areas are not high risk all the time.

Data from this model can act as training data for the other South Asian countries where information was not available for this study. This study identifies the complex relationship between identified FMD risk factors, importantly impact of climate variables for FMD outbreaks. Standardizing the surveillance procedures among countries will improve for capturing occurrences in future studies and regional FMD control.

Further, we emphasize a regional approach of implementing control measures and resource allocation in FMD control. This study facilitate both Sri Lanka and Bangladesh to target risk based surveillance and control activities. Identifying FMD risk areas in the country would help both Sri Lanka and Bangladesh to accomplish zonal compartmentalization in the country, legal frameworks to regulate international animal movements, and requirements of stage 2 of PCP.

## Supporting information

**S1 Fig. Outbreak numbers in different years from Bangladesh, India and Sri Lanka used in the analysis.**
(TIF)

**S1 Table. Source of the different predictors, spatial resolution, and the data collected period considered in this study.**
(DOCX)

**S1 File. Outbreak location data repository.**
(DOCX)

## Author contributions

**Conceptualization:** Umanga Gunasekera, Moh A Alkhamis, Andres Perez.

**Data curation:** Sumathy Puvanendiran, Moumita Das, Pradeep L Kumarawadu, Munawar Sultana, M. Anwar Hossain.

**Formal analysis:** Umanga Gunasekera.

**Funding acquisition:** Jonathan Arzt, Andres Perez.

**Investigation:** Umanga Gunasekera.

**Methodology:** Umanga Gunasekera, Moh A Alkhamis.

**Resources:** Andres Perez.

**Supervision:** Moh A Alkhamis, Andres Perez.

**Validation:** Moh A Alkhamis, Jonathan Arzt.

**Writing – original draft:** Umanga Gunasekera.

**Writing – review & editing:** Umanga Gunasekera, Moh A Alkhamis, Sumathy Puvanendiran, Moumita Das, Pradeep L Kumarawadu, Munawar Sultana, M. Anwar Hossain, Jonathan Arzt, Andres Perez.

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
