## [Decision Letter · Decision Letter 0]

19 Nov 2024

PONE-D-24-43580Ecological niche modeling for surveillance of foot-and-mouth disease in South AsiaPLOS ONE

Dear Dr. Gunasekera,

Thank you for submitting your manuscript to PLOS ONE. After careful consideration, we feel that it has merit but does not fully meet PLOS ONE’s publication criteria as it currently stands. Therefore, we invite you to submit a revised version of the manuscript that addresses the points raised during the review process.

**ACADEMIC EDITOR:** Reviewers recommended major revisions to improve the manuscript. The reviewer's comments are enclosed below. Please carefully consider each point raised by the reviewers.

We look forward to receiving your revised manuscript.

Kind regards,

Nussieba A. Osman, Dr. Med. Vet.

Academic Editor

PLOS ONE

Journal Requirements:

“This project was funded in part by a grant from the USDA:ARS”

3. Thank you for uploading your study's underlying data set. Unfortunately, the repository you have noted in your Data Availability statement does not qualify as an acceptable data repository according to PLOS's standards.

4. We note that Figure 2 in your submission contain map images which may be copyrighted. All PLOS content is published under the Creative Commons Attribution License (CC BY 4.0), which means that the manuscript, images, and Supporting Information files will be freely available online, and any third party is permitted to access, download, copy, distribute, and use these materials in any way, even commercially, with proper attribution. For these reasons, we cannot publish previously copyrighted maps or satellite images created using proprietary data, such as Google software (Google Maps, Street View, and Earth). For more information, see our copyright guidelines: http://journals.plos.org/plosone/s/licenses-and-copyright  

     1. You may seek permission from the original copyright holder of Figure 2 to publish the content specifically under the CC BY 4.0 license. 

Reviewers' comments:

Reviewer's Responses to Questions

**Comments to the Author**

1. Is the manuscript technically sound, and do the data support the conclusions?

Reviewer #1: Partly

Reviewer #2: Yes

2. Has the statistical analysis been performed appropriately and rigorously?

Reviewer #1: Yes

Reviewer #2: Yes

3. Have the authors made all data underlying the findings in their manuscript fully available?

Reviewer #1: Yes

Reviewer #2: Yes

4. Is the manuscript presented in an intelligible fashion and written in standard English?

Reviewer #1: Yes

Reviewer #2: Yes

5. Review Comments to the Author

Reviewer #1: Preliminary observation:

The manuscript entitled “Ecological niche modeling for surveillance of foot-and-mouth disease in South Asia” provides an interesting insight on risk factors associated for FMD in south Asian countries in general and India, Sri Lanka and Bangladesh in particular based on the data available. Of these countries India is in stage 4 of Progressive Control Pathway (PCP) of FMD, whereas other two countries are in PCP 1. Therefore the FMD control in India is at a very dynamic stage. The approach included use of appropriate models to normalize the possibility of outbreaks in different parts of these countries, as under-reporting of the outbreaks is common in the area due to different reasons. While the authors acknowledge the challenge of underreporting outbreaks, they fail to provide a transparent account of how the model compensates for this issue. Clarifying the methodology used to address underreporting is essential for enhancing the study’s credibility. This also included a variety of machine learning algorithms to identify the best fit using different (13) parameters responsible for disease/outbreak and risks associated.

General observations

The English language used in the manuscript is good and easy to understand.

Comments

Minor:

The abbreviation of Ecological niche models (ENM) may be checked as they are written as EMN (lines 64 & 245). A concise explanation of how ENMs function and their relevance to FMD would significantly benefit the reader’s understanding. The figure S1 is missing in the supporting documents.

Major:

• The methodology used for analysis is very appropriate in terms of models (ecological niche model and machine learning tools and different parameters required for risk assessment of FMD. The manuscript mentions the use of random forest, support vector, and gradient boosting algorithms. However, it does not provide information on the selection criteria or performance metrics used for comparison. Also the methodology for handling duplicate outbreak occurrences is inadequately defined, lacking clear criteria for identification and removal. This ambiguity poses a risk of distorting the training dataset used for machine learning models, which could compromise the reliability of the results. A well-defined rationale and systematic approach to duplicate management are crucial to ensure data integrity and enhance model performance.

• The Indian FMD control programme has entered in to a very dynamic stage changing in every round of vaccination every six month. As shown in the manuscript that India contributes to the major niche in the region and remains a major source of viruses in the region, with a changed scenario this may not be true. The factors such as outbreak trend cannot be predicted, until the recent vaccination efforts, and trend of recent outbreaks are used to fit the model from India. This may also affect the prediction in neighbouring countries like Sri Lanka. Interventions during intensive vaccinations are very dynamic and difficult to predict due to changing herd immunity patterns every six months in different parts of the country. The implications of this data dependency should be thoroughly discussed to avoid misleading conclusions.

• Also degree of under reporting varies in different countries taken in the study and also different parts of a country like India needs to be predicted and differentiated (although manuscript discussed this at a point). Higher risk of predictions for India (0.75 as against 0.04 and 0.55 for Sri Lanka and Bangladesh respectively) compared to Sri Lanka and Bangladesh needs to be normalized in view of ongoing efforts, which have reduced the number of outbreaks significantly during the past two years on account of 3-4 rounds of mass vaccinations using a potent FMD vaccine and other measures.

• India is having a higher capability for diagnosis, virus identification and characterization, which sometimes gets reflected in scenario that most of the viruses originate from India (line 288-289) needs to be discussed in the manuscript.

• The analysis primarily considers environmental variables to predict FMD risk but may overlook other crucial epidemiological and socio-economic factors such as herd immunity levels, animal movement patterns, and vaccination compliance. These aspects are likely to be significant drivers of FMD outbreaks and could have strengthened the model’s predictive ability. Adding these non-environmental predictors could yield more comprehensive insights and potentially improve model accuracy. The absence of time-based analysis also limits the model's ability to predict outbreaks dynamically, making it less actionable for real-time decision-making.

Reviewer #2: This paper discusses how ecological niche modelling, coupled with machine learning, can be used to predict the risk of Foot and Mouth Disease (FMD) in the South Asian countries of India, Bangladesh, and Sri Lanka. The paper's main contribution is the development of a spatial risk map for FMD, highlighting areas of high risk in these countries. This information can then be used to inform and improve surveillance and control efforts for the disease in the region. I recommend that this paper be accepted after major revision. Please find full comments from the attached file.

6. PLOS authors have the option to publish the peer review history of their article (what does this mean? ). If published, this will include your full peer review and any attached files.

**Do you want your identity to be public for this peer review?** For information about this choice, including consent withdrawal, please see our Privacy Policy .

Reviewer #1: **Yes: ** Rabindra Prasad Singh

Reviewer #2: No

---

## [Author Response · Author response to Decision Letter 1]

20 Jan 2025

Reviewer 1

Dear Editor and dear Reviewer,

We would like to thank for your detailed review of the manuscript and for the useful comments given. We carefully considered all your comments and thus we addressed them with the appropriate corrections in the text.

Replies and explanations to each comment are listed below.

While the authors acknowledge the challenge of underreporting outbreaks, they fail to provide a transparent account of how the model compensates for this issue. Clarifying the methodology used to address underreporting is essential for enhancing the study’s credibility.

We would like to thank the reviewer for this remark. For different ML algorithms, reported outbreaks are one variable among considered many variables that we use to determine FMD high-risk areas. One objective of this MS was to identify high risk areas based on environmental data and reported outbreaks numbers. Following sentences were added/ modified.

Introduction;

Line 61-66

‘Compared to spatial methods such as kriging (5) and sat scan analysis, which uses only outbreak data location and time (6) to determine FMD risk areas, ecological niche models (ENM) can accommodate and explore many highly correlated and complex risk factors to predict the spatial risk of FMD robustly. ENM use variables, such as temperature and precipitation, along with reported incidences to detect species abundance and predict outbreak occurrence (7)’

Line 75-77

‘Different environmental risk factors such as the relative humidity, temperature are associated with FMD persistence in endemic countries (15). To predict the FMD risk, location of outbreak occurrence data are combined with different environmental predictor’s spatial distribution (16). ‘

Limitations:

Line 429-436

‘There are a number of limitations in our work. The differences in surveillance capacities of the three countries that lead to data inconsistencies are a limitation when comparing the countries for FMD risk.India is having higher diagnostic capabilities compared to both Sri Lanka and Bangladesh. Since we are predicting the suitability for all types of occurrences (reported outbreaks and laboratory confirmed outbreaks via antibody detection) using an ecological model, ecological fallacy may affect the results. To a degree, this is compensated as we regarded only the spatial aspect of data. Our study encourage authorities in all the three countries to improve the quality of FMD surveillance particularly Bangladesh where data was not available from a central source. ’

Minor Comments

The abbreviation of Ecological niche models (ENM) may be checked as they are written as EMN (lines 64 & 245).

Thank you for your observation. This is corrected.

A concise explanation of how ENMs function and their relevance to FMD would significantly benefit the reader’s understanding.

Thank you this suggestion. This is now explained in detail in the introduction.

Line 75-79

‘For FMD and other infectious diseases, there is no straight linear pattern that differentiate infected from the non-infected (15). Machine learning approaches are capable of exploring these nonlinear interactions (16). Different environmental risk factors such as the relative humidity, temperature are associated with FMD persistence in endemic countries (17). To predict the FMD risk, location of outbreak occurrence data are combined with different environmental predictor’s spatial distribution (16). For FMD, ML has been used in studies in Thailand (18), South Africa (19), and China (20). ‘

The figure S1 is missing in the supporting documents.

This figure is uploaded again to the system.

Major comments

The methodology used for analysis is very appropriate in terms of models (ecological niche model and machine learning tools and different parameters required for risk assessment of FMD). The manuscript mentions the use of random forest, support vector, and gradient boosting algorithms. However, it does not provide information on the selection criteria or performance metrics used for comparison.

We would like to thank the reviewer for this remark. In this study we followed the multi algorithm machine learning ensemble pipeline introduced in the ‘How to make more from exposure data? An integrated machine learning pipeline to predict pathogen exposure. J Anim Ecol. 2019 Oct;88(10):1447–61.’ Each ML approach we predicts and identify features differently as explained in line 203-214.

“Random Forrest (RF) and gradient boosting (GB) methods are based on decision trees. Decision trees provide classification and separate paths based on selected variables. The way decision trees are made is different for each method. The RF method is suitable when the data is sparse. In the RF method, variables from the bootstrap data (training data) are randomly allocated in decision trees. Trees are then randomly selected to test data (out-of-bag data) that was not used in creating trees (33) (34). The accuracy of the model is determined by test data. When GB makes decision trees, new trees are scaled, and made based on the errors of previous trees, and the size of the trees is restricted (35). The support vector (SV) machine learning method uses a kernel function to classify data (outbreak vs no outbreak) at a higher dimension space based on a threshold value. The threshold accounts for the bias-variance tradeoff (36)(37). Ten-fold cross-validation was used to estimate model performance and compare different machine learning methods.”

The pipeline used in this MS performs better compared to presence only traditional ecological niche models, as mentioned in line 66-73

‘One of the most commonly used ENM is maximum entropy species distribution. However, this model requires several assumptions such as representative sampling, and considers presence-only data. For the algorithm component, supervised machine learning methods based on decision trees and kernel function are recommended over this method using both presence and absent data (8). Interpretable multi algorithm machine learning models require fewer statistical assumptions, are less sensitive to highly correlated variables and, therefore, overfitting, and can explore none-linear complex relationships between variables (9). ‘

Performance metrics

Performance matrices of each model from k fold cross validation is mentioned in line 215 – 235 and comparison results are shown in the table 1.

Regarding different features, the feature importance plots identify importance of each feature to the model and any feature that is not performing well. Partial dependency plots shows the impact of each feature on model predictions and the interaction plots show the marginal impact (also calculated by Friedman’s H statistics).

Also the methodology for handling duplicate outbreak occurrences is inadequately defined, lacking clear criteria for identification and removal. This ambiguity poses a risk of distorting the training dataset used for machine learning models, which could compromise the reliability of the results. A well-defined rationale and systematic approach to duplicate management are crucial to ensure data integrity and enhance model performance.

We would like to thank the reviewer for this remark. Since models are spatial, all the duplicate outbreaks were removed. Location is considered once for each outbreak. This is explained in line 90

‘to lower the training error of the model, duplicate occurrences of the same geographical location of outbreaks were removed during the considered period. ‘

The granularity of selected outbreak location is now made clear with the added description below. The selected outbreak locations were shared in a data repository.

Line 92-124.

‘In Bangladesh, a systematic FMD outbreak surveillance system is not available. Therefore, it is possible that not all outbreaks are reported during the period. The outbreak data used in this study are from a field study conducted during 2012 to 2021 covering 32 different districts. Outbreak information are collected based on farmer’s notification for serology testing. A total of 481 samples were collected from different outbreak. Sample collection was affected by the 2020-2021 COVID 19 outbreak (18). In this study, we used sample collection location confirmed by laboratory testing affiliated district as the outbreak location. We did not consider the temporal aspect of data. Compared to data coming from Sri Lanka and India, data from Bangladesh is not complete.

For Sri Lanka, officially reported outbreak data were available from the Department of Animal Health, Sri Lanka. The respective veterinarians from nearly 256 veterinary ranges report outbreaks as a part of passive surveillance activities. A district in Sri Lanka consist of multiple veterinary ranges. Outbreak reporting system is a paper based monthly report sent to the head office. Total 369 outbreaks were reported during the period of 2014 to 2022 from different areas of the country. A reported outbreak may have one to many infected animals in an identified farm location. If one or more outbreak was reported at a VS range, a point location of the VS range was considered as positive and recorded for this study. These outbreaks are clinically identified initially with later confirmatory serotype diagnosis.

In India, veterinary authorities conduct both active and passive surveillance for FMD mainly focusing on passive surveillance in nearly 65000 administrative levels. Confirmatory diagnosis is carried out for serotyping in 27 FMD network laboratories and 2 national laboratories. A reported outbreak may have one to many infected animals in an identified farm location. The reporting system is paper based and a monthly report is submitted to the Department of Animal Husbandry. The outbreak location was considered up to the district level (ie if there are one or more outbreaks reported at village level, for the purpose of this analysis, the district was considered positive and a point location of the district was recorded). FMD outbreak data for India (n=429) were available at the district level from a previous study for the years 2009 to 2020 (3).’

The Indian FMD control programme has entered in to a very dynamic stage changing in every round of vaccination every six month. As shown in the manuscript that India contributes to the major niche in the region and remains a major source of viruses in the region, with a changed scenario this may not be true. The factors such as outbreak trend cannot be predicted, until the recent vaccination efforts, and trend of recent outbreaks are used to fit the model from India.

This may also affect the prediction in neighbouring countries like Sri Lanka. Interventions during intensive vaccinations are very dynamic and difficult to predict due to changing herd immunity patterns every six months in different parts of the country. The implications of this data dependency should be thoroughly discussed to avoid misleading conclusions.

Thank you for your observation. We accept that the dynamic nature of FMD occurrence cannot be discussed in this MS since we did not consider the temporal aspect of data with unavailability of outbreak data. However, we try to explain this with following added lines in the discussion Line 334-336,

‘In this model, we considered only environment-related variables for predictions and outbreak data from India until 2020. Therefore, it may not reflect the recent advances the country has made with the FMD control with the vaccination program (49).’

Implications of data dependency now described further in line 92-124 and was added as a limitation in line 439-436.

Also degree of under reporting varies in different countries taken in the study and also different parts of a country like India needs to be predicted and differentiated (although manuscript discussed this at a point). Higher risk of predictions for India (0.75 as against 0.04 and 0.55 for Sri Lanka and Bangladesh respectively) compared to Sri Lanka and Bangladesh needs to be normalized in view of ongoing efforts, which have reduced the number of outbreaks significantly during the past two years on account of 3-4 rounds of mass vaccinations using a potent FMD vaccine and other measures.

Thank you for your observation. The limitation of outbreak data was added and the differences in surveillance program in each country was discussed in further details. It is evident that India has a better surveillance system compared to the other two countries. More details about outbreak reporting systems added in (line 93-117).

Since we are looking at data at a higher scale, ecological fallacy can be associated. Mentioned in line 429-436

‘The differences in surveillance capacities of the three countries that lead to data inconsistencies are a limitation when comparing the countries for FMD risk. India is having higher diagnostic capabilities compared to both Sri Lanka and Bangladesh. Since we are predicting the suitability for all types of occurrences (reported outbreaks and laboratory confirmed outbreaks via antibody detection) using an ecological model, ecological fallacy may affect the results. To a degree, this is compensated as we regarded only the spatial aspect of data. Our study encourage authorities in all the three countries to improve the quality of FMD surveillance particularly Bangladesh where data was not available from a central source.’

India is having a higher capability for diagnosis, virus identification and characterization, which sometimes gets reflected in scenario that most of the viruses originate from India (line 288-289) needs to be discussed in the manuscript.

This sentence was changed to South Asia instead of India in Line 342 and the following sentence was removed.

“Studies link current circulating strains in Sri Lanka (47) (48) and Bangladesh Ind2001BD1 (49) to Ind/2001 that was first identified in India (50). “

The analysis primarily considers environmental variables to predict FMD risk but may overlook other crucial epidemiological and socio-economic factors such as herd immunity levels, animal movement patterns, and vaccination compliance. These aspects are likely to be significant drivers of FMD outbreaks and could have strengthened the model’s predictive ability. Adding these non-environmental predictors could yield more comprehensive insights and potentially improve model accuracy.

This is a good suggestion; unfortunately, no reliable data sources are readily available from any of the countries related to herd immunity, animal movement or vaccination. Even if available, not in the same geographic scale in this model, and incorporating categorical data values repeatedly in a continuous data model, will compromise the model performance.

The absence of time-based analysis also limits the model's ability to predict outbreaks dynamically, making it less actionable for real-time decision-making.

Thank you for your observation. This is added as a limitation line 443-445.

“Here we did not consider the temporal aspect of FMD outbreaks but FMD has a strong temporal component that results in the specific epidemic curve, which indicates this identified high-risk areas are not high risk all the time.”

Reviewer 2

Dear Editor and Reviewer,

We would like to thank for your detailed review of the manuscript and for the useful comments given. We carefully considered all your comments and thus we addressed them with the appropriate corrections in the text.

Replies and explanations to each comment are listed below.

1. Lack of Transparency on Data Comparability: A significant concern due to the lack of detail regarding data comparability between the three countries. While the authors mention their data sources, they do not adequately address potential variations in data collection, reporting standards, and diagnostic confirmation practices. This lack of transparency could misinterpret the risk predictions and undermine the study's overall reliability.

The authors need to provide a more detailed account of the data used for each country, acknowledging potential limitations in comparability. This might involve describing the data collection methodologies used in each country, discussing any inconsistencies in data reporting formats or definitions, and addressing t

---

## [Decision Letter · Decision Letter 1]

27 Feb 2025

Ecological niche modeling for surveillance of foot-and-mouth disease in South Asia

PONE-D-24-43580R1

Dear Dr. Gunasekera,

We’re pleased to inform you that your manuscript has been judged scientifically suitable for publication and will be formally accepted for publication once it meets all outstanding technical requirements.

Kind regards,

Nussieba A. Osman, Dr. Med. Vet.

Academic Editor

PLOS ONE

Reviewers' comments:

Reviewer's Responses to Questions

**Comments to the Author**

1. If the authors have adequately addressed your comments raised in a previous round of review and you feel that this manuscript is now acceptable for publication, you may indicate that here to bypass the “Comments to the Author” section, enter your conflict of interest statement in the “Confidential to Editor” section, and submit your "Accept" recommendation.

Reviewer #1: All comments have been addressed

Reviewer #2: All comments have been addressed

2. Is the manuscript technically sound, and do the data support the conclusions?

Reviewer #1: Yes

Reviewer #2: Yes

3. Has the statistical analysis been performed appropriately and rigorously?

Reviewer #1: Yes

Reviewer #2: Yes

4. Have the authors made all data underlying the findings in their manuscript fully available?

Reviewer #1: Yes

Reviewer #2: Yes

5. Is the manuscript presented in an intelligible fashion and written in standard English?

Reviewer #1: Yes

Reviewer #2: Yes

6. Review Comments to the Author

Reviewer #1: Authors have now addressed all the comments as per observations made. The manuscript may now be accepted for publication.

Reviewer #2: Reviewer Comments and Recommendation for Acceptance

We appreciate the authors’ efforts in revising the manuscript Ecological Niche Modeling for Surveillance of Foot-and-Mouth Disease in South Asia."(PONE-D-24-43580_R1)" The authors have successfully addressed all major concerns, providing clear explanations, methodological improvements, and additional details that enhance the transparency and rigor of the study. Below is a summary of the key reviewer concerns and how the authors have responded effectively.

1. Transparency on Data Comparability

The initial concern was the lack of detailed information regarding how data were collected and compared across India, Bangladesh, and Sri Lanka. The authors have now included comprehensive explanations about data sources, reporting standards, and diagnostic confirmation practices, ensuring greater clarity and reducing potential misinterpretations.

2. Comparability of Predicted Risk Across Countries

The reviewer questioned whether disparities in data availability influenced differences in predicted risk levels rather than actual disease risk. The authors have acknowledged this limitation and further discussed how surveillance capabilities vary across the three countries. Additionally, they have suggested future research to improve data consistency, particularly in Bangladesh, where centralized surveillance is lacking.

3. Potential Data Leakage

Concerns about possible data leakage were raised, as feature selection was performed before splitting the data into training and testing sets. The authors have addressed this by explaining that cross-validation helps mitigate this risk, and they have now included this as a potential limitation to ensure transparency.

4. Class Distribution in Model Evaluation

The absence of an explicit mention of class distribution in model evaluation was noted. The authors have now clarified that a 1:1 case-control ratio was used, ensuring balanced representation of presence and absence cases. Additionally, ROC curves confirmed that no class imbalance was present.

5. Use of the Caret Package for Hyperparameter Tuning

The reviewer suggested that the authors explicitly mention their use of the caret package for hyperparameter tuning. The authors have acknowledged this and have now provided a brief explanation for transparency and reproducibility.

6. Clarification on Variables and Feature Selection

The reviewer requested details on the total number of variables considered and the number removed after correlation filtering. The authors have now included a supplementary table listing all 24 variables, clarified the 9 variables removed due to correlation (>0.9), and confirmed that 15 variables were ultimately used in the analysis.

7. Model Runs and Robustness Evaluation

The reviewer inquired about the number of times the models were run and the robustness of the evaluation process. The authors have now specified that each model underwent 10 runs of 10-fold cross-validation, ensuring reliability and reducing overfitting. This additional detail strengthens the credibility of the model performance results.

Final Recommendation

With these revisions, the manuscript now provides a more transparent, methodologically sound, and well-documented study on ecological niche modeling for foot-and-mouth disease surveillance. The authors have successfully addressed all concerns with thorough explanations, added details, and clear acknowledgments of limitations.

Given the improvements, I recommend accepting this manuscript for publication. This study offers valuable insights into regional FMD risk assessment using machine learning, and the refinements made in response to the reviews significantly enhance its impact and reliability.

Best wishes,

Reviewer 2

7. PLOS authors have the option to publish the peer review history of their article (what does this mean? ). If published, this will include your full peer review and any attached files.

**Do you want your identity to be public for this peer review?** For information about this choice, including consent withdrawal, please see our Privacy Policy .

Reviewer #1: **Yes: ** Rabindra Prasad Singh

Reviewer #2: No

---

## [Editor Report · Acceptance letter]

PONE-D-24-43580R1

PLOS ONE

Dear Dr. Gunasekera,

I'm pleased to inform you that your manuscript has been deemed suitable for publication in PLOS ONE. Congratulations! Your manuscript is now being handed over to our production team.

Kind regards,

on behalf of

Dr. Nussieba A. Osman

Academic Editor

PLOS ONE